# Combination of a Bioceramic Scaffold and Simvastatin Nanoparticles as a Synthetic Alternative to Autologous Bone Grafting

**DOI:** 10.3390/ijms19124099

**Published:** 2018-12-18

**Authors:** Chau-Zen Wang, Yan-Hsiung Wang, Che-Wei Lin, Tien-Ching Lee, Yin-Chih Fu, Mei-Ling Ho, Chih-Kuang Wang

**Affiliations:** 1Orthopedic Research Center, Kaohsiung Medical University, Kaohsiung 807, Taiwan; czwang@kmu.edu.tw (C.-Z.W.); yhwang@cc.kmu.edu.tw (Y.-H.W.); tn916943@gmail.com (T.-C.L.); microfu@gmail.com (Y.-C.F.); homelin@kmu.edu.tw (M.-L.H.); 2Department of Physiology, Kaohsiung Medical University, Kaohsiung 807, Taiwan; 3Graduate Institute of Medicine, Kaohsiung Medical University, Kaohsiung 807, Taiwan; 4Department of Medical Research, Kaohsiung Medical University Hospital, Kaohsiung 807, Taiwan; 5School of Dentistry, Kaohsiung Medical University, Kaohsiung 807, Taiwan; 6Department of Medicinal and Applied Chemistry, Kaohsiung Medical University, Kaohsiung 807, Taiwan; j23g21d21o32@gmail.com; 7Department of Orthopedics, Kaohsiung Medical University Hospital, Kaohsiung Medical University, Kaohsiung 807, Taiwan; 8Department of Orthopedics, Kaohsiung Municipal Hsiao-Kang Hospital, Kaohsiung Medical University, Kaohsiung 812, Taiwan; 9Department of Chemistry, National Sun Yat-Sen University, Kaohsiung 804, Taiwan

**Keywords:** combination, bioceramic, nanoparticles, simvastatin, sustained release, scaffold

## Abstract

The fragile nature of porous bioceramic substitutes cannot match the toughness of bone, which limits the use of these materials in clinical load-bearing applications. Statins can enhance bone healing, but it could show rhabdomyolysis/inflammatory response after overdosing. In this study, the drug-containing bone grafts were developed from poly(lactic acid-co-glycolic acid)-polyethylene glycol (PLGA-PEG) nanoparticles encapsulating simvastatin (SIM) (SIM-PP NPs) loaded within an appropriately mechanical bioceramic scaffold (BC). The combination bone graft provides dual functions of osteoconduction and osteoinduction. The mechanical properties of the bioceramic are enhanced mainly based on the admixture of a combustible reverse-negative thermoresponsive hydrogel (poly(*N*-isopropylacrylamide base). We showed that SIM-PP NPs can increase the activity of alkaline phosphatase and osteogenic differentiation of bone marrow stem cells. To verify the bone-healing efficacy of this drug-containing bone grafts, a nonunion radial endochondral ossification bone defect rabbit model (*N* = 3/group) and a nonunion calvarial intramembranous defect Sprague Dawley (SD) rat model (*N* = 5/group) were used. The results indicated that SIM-PP NPs combined with BC can improve the healing of nonunion bone defects of the radial bone and calvarial bone. Therefore, the BC containing SIM-PP NPs may be appropriate for clinical use as a synthetic alternative to autologous bone grafting that can overcome the problem of determining the clinical dosage of simvastatin drugs to promote bone healing.

## 1. Introduction

Bones comprise the majority of both volume and weight in the human body, and their most important function is to allow the body’s action and support the body structure. The most common bone defects in clinical treatment include serious bone fractures, nonunion bone fractures, bone defects caused by osteomyelitis, postoperative defects caused by bone tumors, vertebral body collapse, and other acetabular bone defects from artificial joint reconstruction [1].

Bone transplantation remains a common treatment. Surgical techniques currently used to treat bone defects rely on different alternatives, including homologous bone grafts provided by a musculoskeletal tissue bank (allograft), autologous vascularized bone grafts, prostheses, or heterologous bone grafts (xenograft), each of which has specific advantages, complications, and drawbacks [1]. However, autologous bone grafts remain the gold standard in bone repair and regeneration because of the osteogenicity, osteoinductivity, and osteoconductivity associated with this technique [2]. However, the quantity of bone available is limited, and the harvesting procedure requires a second surgical site that result in severe complications [3,4]. In addition, autologous bone grafts may be resorbed too rapidly as they can be degraded before bone completed healing [5].

We recognized that an ideal bone formation strategy is for artificially synthesized biomaterials to be biocompatible, bioresorbable, osteoinductive, osteoconductive, easy to use and integrable into the surrounding host bone [6,7]. Osteoconductive synthetic bone substitutes are the most widely used alternative material. Many synthetic bone graft materials are available as alternatives to autogenous bone, including calcium sulfate, bioactive glasses and calcium phosphates (tricalcium phosphate, TCP; calcium hydroxyapatite, HA; and biphasic calcium phosphate, BCP) [8].

Some bone morphogenetic proteins (BMPs) have been reported to stimulate the proliferation and differentiation of osteogenic cells in vitro and in vivo [9]. Among them, BMP-2 and BMP-7 have been clinically applied for nonunion and fractures [10,11]. However, local BMP delivery systems either require a high concentration bolus dose or sustained delivery for bone tissue engineering due to their short half-lives [12]. Some reports have shown that high BMP concentrations are related with increased osteoclastic activity and bone resorption [13,14]. In addition, small-molecule drugs can also promote the function of bone growth. For example, Mundy et al. showed that statins can stimulate BMP-2 gene expression in osteoblasts [15,16]. Several in vitro studies indicated that statins can increase the expression of osteogenic marker genes other than BMP-2, including alkaline phosphatase, runt-related transcription factor 2 (RUNX-2), osteopontin, and osteocalcin, in osteoblastic cells and bone marrow-derived cells [17,18]. We demonstrated an effect of controlled release of simvastatin (SIM) from poly (lactic-co-glycolic acid)/hydroxyapatite) (PLGA/HAp) microspheres in vitro and improved bone healing of necrotic bone grafts in vivo [19,20]. We concluded that calcium phosphate bioceramics combined with simvastatin (SIM) microcarriers can stimulate osteogenic differentiation of stem cells in vitro and bone induction in vivo. The use of a bone substitute with local release of SIM microcarriers to support bone growth provides a wealth of potential clinical use for the treatment of critical bone defects and fractures.

However, some studies have shown side effects of statins in vivo, such as rhabdomyolysis or inflammatory response after overdosing [21]. Therefore, if the patient has a severe bone defect and a bioceramic bone graft is used clinically with simvastatin (SIM) carriers to enhance bone generation, the physician will still need to determine the dose of SIM/PLGA microspheres and the sites of administration at the bone defect. The dose depends on the size of the defect and how well the microspheres are placed and is currently a very troubling problem for clinicians. Therefore, the goal of this study is to develop a combination product of drug-containing bone grafts based on a porous bioceramic containing simvastatin nanoparticles to enhance bone healing and overcome the problem of determining the dosage of simvastatin drugs during surgery to promote bone growth.

## 2. Results

### 2.1. Synthesis of the PLGA-PEG (PP) Copolymer and the Properties of SIM/PP Nanoparticles

The segment copolymer of H_2_N-PEG-OMe can directly conjugate to PLGA-COOH through an amide linkage to form the PLGA-PEG-OMe copolymer structure (Appendix A), and ^1^H NMR spectra verified the PLGA-PEG-OMe structure (Appendix A), with a signal peak at 1.562 ppm representing PLA methyl protons (PLA CH_3_) and a peak at 4.667–4.817 ppm representing PGA methylene protons (CH_2_). The peak at 5.219 ppm, related to the CH–CH_3_ from lactic acid, and the peak at 3.641 ppm, representing polyethylene glycol (PEG) ether linkages, both confirmed the synthesis of the copolymer.

The average NPs size for the PLGA-PEG-OMe copolymers was 120.3 ± 8.5 nm and the average zeta potentials of the SIM/PP NPs were −32.7 mV, as indicated in Table 1, similar to previous results [20]. A schematic illustration of the SIM/PP NP composition construction and a photographic image of the SIM/PP NPs colloid is shown in Appendix A. In addition, the encapsulation efficiency and loading efficiency of SIM in the SIM/PP NPs were 33.6 ± 3.5% and 3.2% according to the HPLC method, as also shown in Table 1.

### 2.2. Synthesis of the p(NiPAAm-MAA) Copolymer

In order to observe the apparent ^1^H-NMR signals of MAA in p(99NiPAAm-1%MAA) copolymer. This p(97% NiPAAm-3%MAA) copolymer was used to enhance the MAA units of the copolymer for ^1^H-NMR experiment. The ^1^H-NMR results for the p(97%NiPAAm-3%MAA) copolymer (Appendix A) indicate successful polymerization of NiPAAm and MAA. These data contain two broad peaks (-CH-CH_2_-) at 1.4 and 1.9 ppm, a peak (-CH_3_) due to the isopropyl group at 1.0 ppm, and a peak (-NH-CH-) at 3.8 ppm that was attributed to the hydrogen proton of the p(NiPAAm) units of the chain. The minor MAA units of the copolymer show that the intensity of the methyl group was significantly enhanced due to the introduction of the methyl group of p(NiPAAm) (chemical shift at 1.2 ppm). However, the double-bond peaks of the NiPAAm monomer at 5.7 ppm and 6.1 ppm have disappeared [8].

### 2.3. Characterization of the Bioceramics

In this study, the p(99%NiPAAm-1%MAA) hydrogels can also act as a suitable binder template for preparing the two plastic molds. As shown in Appendix A, one was a round disk mold (ϕ 5 mm; h 0.7 mm), and the other was a hollow cylinder mold (ϕ_1_ 3.5 mm; ϕ_2_ 1.5 mm; h 10 mm). The appearance of the porous bioceramics from the two molds after sintering at 1200 °C for 2 h is shown in Figure 1(a1,b1). The morphology of the SIM-PP NPs loaded within the porous bioceramic hollow cylinder after sintering at 1200 °C for 2 h is shown in the SEM images in Appendix A. Those macropores (100–600 μm) can be obtained from the air bubbles generated during the mixing process. In addition, the minimum crevices (<5–100 μm) can be attributed to a lack of sintering densification. However, the liquid permeability results in Appendix A show that the red ink permeated through the whole body of the porous bioceramic by capillarity.

Appendix A shows the XRD patterns of the bioceramics after sintering at 1200 °C for 2 h. The XRD peaks of all of the diffraction patterns agree well with those of standard HAp and β-TCP in the powder diffraction file (Card NO. 9–432 and 9–169). A phase calibration curve of the XRD spectrum was prepared using standard HAp and β-TCP, and 50% HAp/50% β-TCP phase weight ratios are reported in Table 2.

### 2.4. Release Profile and Bioactivity of act-SIM from SIM-PP NPs Loaded within the Bioceramics

The total amount of simvastatin each disk bioceramic carried was 2.5 μg (SIM-PP NP 2.5/BC_disk_) or 5.0 μg (SIM-PP NP 5/BC_disk_). The daily release profiles of the active form of simvastatin (act-SIM) from SIM-PP NPs loaded within the round disk bioceramic samples (Figure 1(a2)) and the average percentages of the act-SIM cumulative release amounts (Figure 1(a3)) were measured by HPLC at 37 °C for 10 days. The results indicated that there was no large initial burst of release from either of these samples, as approximately 20% was released on the first day. However, an average daily release of 0.51–0.30 µM act-SIM from the SIM-PP NP 2.5/BC_disk_ was observed for 5 days, and an average daily release of 0.79–0.25 µM act-SIM from the SIM-PP NP 5/BC_disk_ was observed for 8 days. The cumulative released act-SIM was near 80% of the total concentration after 5 and 10 days.

The total amount of simvastatin each cylinder bioceramic carried was 5.0 μg (SIM-PP NP 5/BC_cylinder_) or 10.0 μg (SIM-PP NP 10/BC_cylinder_). The daily release profiles of the active form of simvastatin (act-SIM) from SIM-PP NPs loaded within the hollow cylinder bioceramic samples (Figure 1(b2)) and the average percentages of the act-SIM cumulative release amounts (Figure 1(b3)) were also measured by HPLC at 37 °C for 8 days. In contrast to the previous results, an initial release burst was observed from both of these cylinder bioceramic samples, with more than 40–50% release on the first day. The average daily release for 3 days was 1.31–0.14 µM act-SIM from the SIM-PP NP 5/BC_cylinder_ and 3.58–0.65 µM act-SIM from the SIM-PP NP 10/BC_cylinder_. The cumulative released act-SIM amounts were near 66% and 82% of the total concentration after 3 days of release, respectively.

To test the bone differentiation bioactivity of the act-SIM released from the SIM-PP NP 5/BC_disk_, we treated BMSCs with a specific concentration. The BMSCs cultured in osteoinduction medium (OIM) without SIM were used as a control, and the 0.5 μM SIM group was used as the experimental group in this study. To confirm the biological effects of act-SIM released from the SIM-PP NP 5/BC_disk_ on BMSC osteogenesis, we investigated the effects of act-SIM on the induction of ALP activity in BMSCs. Our results showed that ALP activity was significantly increased in cells treated with 0.5 μM act-SIM, and we theorized that 0.5 μM act-SIM released from the SIM-PP NP 5/BC_disk_ would produce similar results (Figure 2).

On the other hand, ALP activity did not appear to increase in cells treated with 0.5 μM act-SIM and 0.5 μM act-SIM from the SIM-PP NP 5/BC_disk_ when cultured in OIM for 3 additional days. However, the act-SIM released from the SIM-PP NP 5/BC_disk_ had good osteogenesis effects when cultured in OIM for 5 additional days.

### 2.5. In Vivo Experiments with Calvarial Defects in Rats

#### 2.5.1. Micro-CT Analysis

To investigate whether the bioceramics combined with SIM-PP NPs had therapeutic effects on the healing of intramembranous bone defects, we used the nonunion calvarial defect (diameter 5 mm) model in rats (Figure 3a). Micro-CT analysis (Figure 3b) of the calvarial defect-only group confirmed the nonunion of the calvarial defect 8 weeks after surgery, with some slight new bone formation on the inside and edge of the defect. In both the SIM-PP 2.5/BC_disk_ group and the SIM-PP 5/BC_disk_ group, the bioceramic scaffolds remained in the calvarial defect sites 8 weeks after surgery. To evaluate the new bone formation inside the bioceramic scaffold, we used the Hounsfield unit (HU) calibration of micro-CT images, with the density distribution of bone revealed by green coloring and the bioceramic scaffold revealed by purple coloring. Appendix A shows that there was obviously more new bone formation inside the bioceramic scaffold in the SIM-PP 5/BC_disk_ group than in the other groups after 8 weeks of healing.

#### 2.5.2. Histological Analysis and Bone Tissue Callus Area Calculation

We confirmed the new bone formation in the calvarial defect by H&E staining in Figure 4a–d. The results of the histomorphological analysis revealed an ingrowth of osteoblasts in the interior of the bioceramic, and obvious bone healing and a bony bridge were observed in the SIM-PP 2.5/BC_disk_ group and the SIM-PP 5/BC_disk_ group (Figure 4c,d). The quantification of new bone formation also showed that both the SIM-PP 2.5/BC_disk_ group and the SIM-PP 5/BC_disk_ group had significantly greater new bone formation inside the bioceramics than the controls, in a dose-dependent manner (Figure 4e).

### 2.6. In Vivo Experiments of Large Radial Bone Defect in Rabbits

#### 2.6.1. Soft X-ray Observation

The radial bone defects were treated with a bioceramic bone graft alone (BC_cylinder_ group) or with SIM/PP-containing bioceramic bone grafts (SIM-PP 5/BC_cylinder_ and SIM-PP 10/BC_cylinder_ groups). To investigate whether the bioceramics containing SIM-PP NPs had a therapeutic effect on the healing of endochondral bone defects, we used the nonunion rabbit radial bone defect model (Figure 5a). Healing of the radial bone was evaluated every two weeks using X-ray analysis, and the results showed that weekly bone growth and bony callus formation were superior in the SIM-PP 10/BC_cylinder_ group compared to the other two groups from 6 weeks to 10 weeks (Figure 5b).

#### 2.6.2. Histological Analysis and Bone Tissue Callus Area Calculation

As X-ray analysis cannot distinguish between the BC and new bone formation in the defect area, we next used H&E analysis to observe and quantify the detailed bone growth in the defect area. The H&E staining results showed bridging of the defect area with ingrowth of osteoblasts invading into the bioceramic interior and significant bone-healing effects in both the SIM-PP 5/BC_cylinder_ and SIM-PP 10/BC_cylinder_ groups. These effects were superior in the SIM-PP 10/BC_cylinder_ group when compared to the other two groups at 10 weeks (Figure 6a). Quantification of the H&E results showed that the SIM-PP 10/BC_cylinder_ group had the highest new bone formation (Figure 6c). New blood vessel formation in the defect area was needed for osteogenic precursor cells to invade into the BC interior. IHC staining of the angiogenesis marker von Willebrand factor (VWF) also revealed higher VWF staining in both the SIM-PP 5/BC_cylinder_ and SIM-PP 10/BC_cylinder_ groups compared to the BC_cylinder_ group at 10 weeks (Figure 6b). These results confirmed that the rabbit groups treated with bioceramics containing SIM-PP NPs exhibited repair of nonunion radial bone defects.

## 3. Discussion

Previously, we have demonstrated that simvastatin can express osteogenic marker genes and promote bone growth in vivo [19,20,22]. Furthermore, we also demonstrated an effect of controlled release of simvastatin (SIM) from poly(lactic-co-glycolic acid)/hydroxyapatite) (PLGA/HAp) microspheres in vitro and improved bone healing of necrotic bone grafts in vivo [19]. However, the excessive use of simvastatin will have rhabdomyolysis or inflammatory response side effects. Therefore, the physician will still need to determine the dose of SIM/PLGA microspheres on the sites of administration at the bone defect. It is still troubling problem for clinicians. Therefore, the goal of this study is to develop a combination product of drug-containing bone grafts based on a porous bioceramic containing simvastatin nanoparticles to enhance bone healing and overcome the problem of determining the dosage of simvastatin drugs during surgery to promote bone growth. This study demonstrated that the combination product of a porous BC scaffold combined with SIM-PP NPs can improve the healing of bone defects in a rabbit radial endochondral ossification bone defect model and an SD rat calvarial intramembranous defect model.

### 3.1. Synthesis of the PP Copolymer and the Properties of SIM/PP Nanoparticles

The methoxy-functionalized diblock copolymer of PLGA-PEG-OMe (PP) was successfully synthesized by conjugating the heterofun ctional PEG with a terminal amine and carboxylic acid functional group to PLGA-COOH using standard carbodiimide/NHS-mediated chemistry, as in our previous study [20]. The other words, all of the segment copolymers between H_2_N-PEG-OMe and PLGA-COOH can be through an amide linkage to form the PLGA-PEG-OMe copolymer structure by ^1^H NMR spectra, both confirmed the synthesis of the copolymer.

Because of the PLGA-PEG-OMe (PP) is an amphoteric copolymer and can be expected to carry simvastatin, a hydrophobic drug, in PP NPs. Therefore, to prepare the SIM/PP NPs, 25 mg of the copolymers was dissolved in 1 mL of DMSO solvent, and then 2 mL of distilled water was added to precipitate the copolymer NPs. The average NP size for the PLGA-PEG-OMe copolymers was 120.3 ± 8.5 nm, which can prompt these nanoparticles to penetrate the micropores of the bioceramic structures. In addition, the average zeta potentials of the SIM/PP NPs were −32.7 mV and the negative charge will be attracted to the positively charged portion of calcium phosphate.

The active carboxylate form of simvastatin (act-SIM) has higher solubility in aqueous solution than the lactone form of simvastatin, and only the carboxylate form of simvastatin can be detected in our HPLC condition [20]. However, we detected the act-SIM amounts from SIM/PP NPs by treating them with 500 μL of ethanol to dissolve SIM and with 800 µL of 0.1 M NaOH to completely break down the PP copolymers (refer to the methods section for the detailed method). Whatever, the encapsulation efficiency and loading efficiency of SIM in the SIM/PP NPs were 33.6 ± 3.5% and 3.2% in this study according to the HPLC method.

### 3.2. Characterization of the Bioceramics Fabricated Using p(99%NiPAAm-1%MAA) Hydrogel

The reverse negative thermo-responsive hydrogel of the p(99%NiPAAm-1%MAA) has homogeneous solution in lower critical solution temperature (LCST) but it can be uniform shrinkage and dehydration when the temperature is more than 32 °C and lose most of water volume. As we previously reported [8], the reverse negative thermos-responsive hydrogel can be regarded as a cold isostatic press (CIP) method to form a compact green body of ceramic. The slurry of HAp/β-TCP/p(99%NiPAAm-1%MAA) can automatically shrink and compress the bioceramic powders to enhance its mechanical properties during sintering densification [8]. Therefore, the p(99%NiPAAm-1%MAA) hydrogel can also act as a suitable binder template for preparing the porous bioceramics in two plastic molds and sintering at 1200 °C for 2 h, which one was a round disk mold (ϕ 5 mm; h 0.7 mm), and the other was a hollow cylinder mold (ϕ_1_ 3.5 mm; ϕ_2_ 1.5 mm; h 10 mm).

Porous bioceramic scaffolds play a key important role in providing a three-dimensional (3D) environment for angiogenesis, the attachment and proliferation of cells in bone tissue engineering. Many kinds of porous bioceramic fabrication methods, including particulate/salt leaching, the polymer burning-out method, gas foaming, freeze drying, fiber bonding, emulsification, phase and separation/inversion, are difficult to control the pore shape, architecture, porosity, or interconnectivity of the bioceramic scaffolds. However, the mechanical strength is often too low to match the compressive strength of cancellous bone (approximately 4–12 MPa) [8]. Increasing the compressive strength of scaffolds without decreasing the porosity and pore size is a challenge for researchers.

These average porosity (56.29 ± 0.55%) and bulk density (1.5 ± 0.01 g/cm^3^) of our sintered bioceramics were obtained. However, a range of pore sizes was easy to obtain for these porous bioceramics that were fabricated from 10% p(NiPAAm-MAA) hydrogel solutions. The advantage of high-porosity scaffolds for achieving a good cellular distribution has been reported, and it is critical to select the correct pore size [23]. However, those macropores (100–600 μm) can be obtained from the air bubbles generated during the mixing process. Because air bubbles are entrapped when viscous fluids are stirred using mechanical force, the plastic viscosity decreases with increasing air content. Some reports [24] have indicated that pores between 100 and 1000 µm play an important role in cellular and bone ingrowth as essential for blood flow. This finding indicates that the porous bioceramic products fabricated using this new method have a suitable pore size range (100–600 μm) for the growth of bone and blood vessels. In addition, the minimum crevices (<5–100 μm) can be attributed to a lack of sintering densification.

Current methods for introducing porosity into a bioceramic material are mainly based on the admixture of the combustible reverse thermo-responsive hydrogel, which burns away during sintering, and on the air bubbles generated during the mixing process, which leave free spaces in the resulting object. However, there was no obvious interconnection between the closed macropores. Therefore, the liquid permeability tests helped clarify whether these crevices had liquid permeability and whether the large pores had interconnection channels. The liquid permeability results also show that the red ink permeated through the whole body of the porous bioceramic by capillarity. Therefore, a porous bioceramic fabricated using this type of new method will allow the movement of nutrients or metabolites when implanted in animals. In addition to, the XRD peaks of these bioceramics showed about 50% HAp/50% β-TCP phase weight ratios. This means that the more highly soluble β-TCP should be easily dissolved to help produce interconnecting pores.

### 3.3. Release Profile and Bioactivity of Act-SIM from SIM-PP NPs Loaded within the Bioceramics

Because the proposed application of the bioceramic scaffold loaded with simvastatin nanoparticles is for sustained drug delivery for bone regeneration, the drug release mechanisms are as important as the drug carrier polymer formulation. In general, the drug release rate depends on the solubility, diffusion, and biodegradation of the matrix materials [25]. A rapid initial release or burst is mainly attributed to weakly bound or adsorbed drug on the large surfaces of the nanoparticles. Slower release occurs by diffusion or erosion of the matrix under sink conditions. If the diffusion of a drug is more rapid than the matrix erosion, then the mechanism of release is largely controlled by a diffusion process.

There was no large initial burst of act-SIM release from either of these SIM-PP NPs/BC_disk_ samples, as approximately 20% was released on the first day. Because of an average daily release of 0.51–0.30 µM act-SIM from the SIM-PP NP 2.5/BC_disk_ was observed for 5 days, and an average daily release of 0.79–0.25 µM act-SIM from the SIM-PP NP 5/BC_disk_ was observed for 8 days. The reason is that amounts less than 0.1 µM act-SIM could not be measured because of the detection limit of our HPLC system. Therefore, we could not detect any act-SIM released from SIM-PP NP 2.5/BC_disk_ before 5 days. In addition, a higher initial release burst was observed from both of these cylinder bioceramic samples, with more than 40–50% release on the first day. Because the volume of the hollow cylinder bioceramic is 5.7 times greater than that of the round disk bioceramic, a larger amount of basic ions from the bioceramic can accelerate the degradation of the PLGA nanoparticles [26], and its slightly alkaline ions may lead to an acceleration of the open-loop structure of active simvastatin production and release [26].

### 3.4. Bioactivity of Act-SIM from SIM-PP NPs/BCs

To test the bone differentiation bioactivity of the act-SIM released from the SIM-PP NP 5/BC_disk_, we treated BMSCs with a specific concentration. Our previous study demonstrated that 0.5 μM SIM was an effective dose for inducing BMSC osteogenesis [19]. Moreover, our previous results have demonstrated that both bioceramic scaffolds [8] and SIM-PP NPs [20] have no cytotoxicity in BMSCs. Therefore, BMSCs cultured in osteoinduction medium (OIM) without SIM were used as a control, and the 0.5 μM SIM group was used as the experimental group in this study.

### 3.5. In Vivo Experiments with Calvarial Defects in Rats and Large Radial Bone Defect in Rabbits

Bone fracture healing is a highly complex and ordered process that involves both intramembranous and endochondral ossification with three overlapping stages: the early inflammatory stage, which includes hematoma formation followed by initial fibrocartilage formation (soft callus); the subsequent regenerative stage, with the removal of fibrocartilage followed by the formation of a hard callus of woven bone; and finally a bone remodeling stage, with remodeling of the woven bone into a lamellar bone structure. Bone fractures can heal spontaneously, but in severe traumatic injury and pathological fractures, approximately 5–10% of fractures become delayed union or nonunion defects that remain challenging for orthopedic surgical correction [19,27]. The calvarial defect model has been applied extensively in basic and applied research and permits the assessment of bone regeneration, especially the intramembranous ossification process in physiological and pathological conditions and the regeneration of craniofacial defects [28,29]. Fractures result in significant morbidity, particularly in light of the increasing frequency of fractures associated with osteoporosis and the elderly. However, the calvarial bone defect model does not allow the assessment of the bone regeneration response to physiological biomechanical loading at a load-bearing location, such as a femur. The femoral fracture defect model permits the assessment of bone regeneration, especially the endochondral ossification process, in response to biomechanical loading under pathological conditions [30,31]. This study is the first to demonstrate that bioceramics containing SIM-PP NPs can have therapeutic effects on the repair of both nonunion calvarial defects in rats and radial bone defects in rabbits.

## 4. Materials and Methods

### 4.1. Materials

Hydroxyapatite (HAp) (CAPTAL^®^S, Plasma Biotal Limited, Derbyshire, UK) and β-TCP (Sigma-Aldrich Inc., St. Louis, MO, USA) were used as the main raw materials in this study. *N*-(3-Dimethylaminopropyl)-*N*’-ethylcarbodiimide hydrochloride (EDC), nuclear magnetic resonance (NMR) d-solvents, tetramethylsilane (TMS), calcium hydride and poly(lactic-co-glycolic acid) (PLGA 50/50, Mw 50,000–75,000, no. 430447) were purchased from Sigma-Aldrich Co. H_2_N-PEG-OMe (Mw 3400) was purchased from Laysan Bio., Inc. (Arab, AL, USA); *N*,*N*’-dicyclohexylcarbodiimide (DCC) and *N*-hydroxysulfosuccinimide (Sulfo-NHS) were purchased from Acros Organics (NJ, USA); and trifluoroacetic acid, hydrochloric acid, and triethylamine were purchased from Riedel-de Haen^®^ (Sigma-Aldrich Inc.). Reagent-grade *N*-isopropylacrylamide (NiPAAm), methacrylic acid (MAA), the activator *N*,*N*,*N*’,*N*’-tetramethylethylenediamine (TEMED), and the initiator ammonium persulfate (APS) were purchased from Sigma-Aldrich, Inc. Sodium hydroxide and magnesium sulfate anhydrous powder were purchased from Sodium hydroxide and magnesium sulfate anhydrous powder were purchased from Showa PK Co. (Tokyo, Japan). Simvastatin was purchased from Merck Inc. (Kenilworth, USA). All other solvents were purchased from TEDIA (Fairfield, OH, USA) or J. T. Baker (Phillipsburg, NJ, USA). The other chemicals were of analytical/reagent grade and were used without further purification.

### 4.2. Synthesis and Characterization of the PLGA-PEG-OMe Copolymers

The synthesis methods of PLGA-PEG-OMe (PP) copolymer were as described previously [20]. Briefly, the 300 mg (0.025 mmol) of PLGA (50/50) with terminal carboxylic acid was activated with 8 mg (0.039 mmol) of DCC in 3 mL of anhydrous dimethylformamide (DMF) under a nitrogen atmosphere for 4 h (a PLGA: DCC mole ratio of 1:1.5). In another flask, 100 mg (~0.029 mmol) of NH_2_-PEG-OMe was dissolved in 2 mL of anhydrous DMF with 0.1 mL of triethylamine (TEA) for 4 h and then was added to the DCC-activated carboxylic acid end group of PLGA solution in a dropwise manner (PLGA: NH_2_-PEG-OMe mole ratio of 1:1.2). The reaction mixture was still stirred under a nitrogen atmosphere for overnight. The solution was dialyzed (molecular weight cut-off; MWCO 1000 Da) against distilled water for 12 h to remove the unreacted NH_2_-PEG-OMe, and then freeze-dry. The PP copolymer product was identified by infrared (IR) and NMR spectroscopy. Infrared spectra (IR) were obtained using a Perkin-Elmer System 2000 FT-IR spectrophotometer (Waltham, MA, USA). ^1^H-NMR spectra were recorded on a Varian Gemini-200 spectrometer (Palo Alto, CA, USA) using TMS as an internal standard. The chemical shifts are given in *δ* (ppm), and the coupling constants are given in Hz.

### 4.3. Formulation and Characterization of the SIM-PP Nanoparticles

This technique for encapsulating simvastatin (SIM) in PLGA-PEG-OMe nanoparticles (PP NPs) to form SIM-PP NPs by precipitation-solvent evaporation was also previously methods [20]. Which was as follows, 25 mg of PP copolymers and 2.5 mg of SIM were dissolved in 1 mL of dimethyl sulfoxide (DMSO) completely. NPs were created by adding the polymer solution to 2 mL of distilled water using the solvent displacement method, because of DMSO is miscible with water. Resulting NPs suspensions were stirred uncovered for 10 min at room temperature, and then dialyzed (MWCO 1000 Da) against distilled water for 2 days to remove the DMSO [20,32]. The particle size distributions of these NPs colloid measured by a Zetasizer 1000 zeta potential instrument (Zetasizer 3000-HSA & Mastersizer M-2000 P-III, Malvern, UK).

### 4.4. Encapsulation Efficiency and Loading Efficiency of SIM-PP NPs

The carboxylate active form of simvastatin (act-SIM) will be released after SIM-PP NPs degradation and be active in cells. The act-SIM form converted from the lactone form of simvastatin was according to the protocol developed by Merck & Co. (Kenilworth, NJ, USA) and prepared it as followed by Keyomarsi [33]. Briefly, 40 mg of SIM was dissolved in 1 mL of alcohol and then mixed with 1.5 mL of 0.1 N NaOH. The act-SIM stock solution was heated at 50 °C for 2 h, and the pH neutralized with 1 N HCl (pH = 7.4) to prepare a 10 mM stock. Calibration curves of act-SIM obtained over a concentration range of 10^−4^ to 10^−8^ µM. Then, 300 µL of the SIM-PP NPs colloid (after 4 days of dialysis) was treated with 500 µL of ethanol to dissolve the SIM, followed by treatment with 800 µL of 0.1 M NaOH to completely break down the PP copolymers. The resultant solution was subsequently neutralized with 0.1 M HCl (α µL). Finally, 80 µL of the clear solution was used for HPLC analysis (Lachrom Elite high-performance liquid chromatograph, Hitachi, Kaigan Minato-ku, Tokyo, Japan). The mobile phase consisted of 1.0% acetic acid in MeCN/CH_3_OH/H_2_O (45/30/25), and the flow rate was set at 1 mL/min. The column effluent detected at 227 nm with a UV/VIS detector. Separation was achieved using an Inertsil^®^ C18 (250 mm × 4.6 mm, 5 µm) analytical column connected to an Inertsil^®^ C18 (50 mm × 4.6 mm, 5 µm) guard column. The column temperature was 40 °C. The moles of SIM within the PP NPs calculated from the concentration of sample × volume of sample ((300 + 500 + 800 + α) µL) × molecular mass of act-SIM. The encapsulation efficiency was calculated by Formula (1), and the loading efficiency was calculated by Formula (2):(1)$$\mathrm{Encapsulation}\;\mathrm{Efficiency}\;\left(\mathrm{EE}\right)\;\mathrm{percentage}=\frac{\mathrm{Mass}\;\mathrm{of}\;\mathrm{drug}\;\mathrm{within}\;\mathrm{PP}\;\mathrm{NPs}}{\mathrm{Mass}\;\mathrm{of}\;\mathrm{loading}\;\mathrm{drug}}\times 100\%$$
(2)$$\mathrm{LoadingEfficiency}\;(\mathrm{LE})\;\mathrm{percentage}=\frac{\mathrm{Mass}\;\mathrm{of}\;\mathrm{drug}\;\mathrm{within}\;\mathrm{PP}\;\mathrm{NPs}}{\mathrm{Mass}\;\mathrm{of}\;\mathrm{total}\;\mathrm{PP}\;\mathrm{polymer}}\times 100\%$$

### 4.5. Preparation and Characterization of the Disk Bioceramics and Hollow Cylinder Bioceramics

The negative thermal-responsive hydrogel of poly[(*N*-isopropylacrylamide)-*co*-(methacrylic acid)] (p(NiPAAm-MAA)) was polymerized by free-radical cross-linking in a solution of 99 mole% NiPAAm and 1 mole% MAA (PNM99:1); the main parameters were as showed previously [8]. Briefly, the target mass of 5 g of NiPAAm and 40 μL of MAA were added to 25 mL of deionized water in a 100 mL beaker and stirred for 1 h. The NiPPAm/MAA solution was deoxygenated by bubbling with nitrogen when it heated to 50 °C for 40 min. Then, 500 μL of TEMED and 0.05 g of APS added to this beaker to initiate addition polymerization. However, the addition polymerization reaction was kept at 50 °C under a nitrogen atmosphere for 24 h with constant stirring. The final PNM99:1 hydrogel formed and then purified by membrane dialysis against distilled deionized water using a Spectra/Pros membrane with a MWCO 1000 Da. The PNM99:1 copolymer was then lyophilized to form a white powder. The purified PNM99:1 copolymer was also confirmed by ^1^H-NMR spectra, which were recorded on a nuclear magnetic resonance spectrometer (Varian Gemini-200, Burladingen, Germany), and the sample was dissolved in D_2_O.

Then, bioceramic scaffolds prepared as previously described [8]. The main procedures were as followed, the bioceramics were prepared using aqueous solutions of 2 mL of 10 *wt*% PNM99:1 hydrogel mixed with 2 g of HAp/β-TCP powders to form a ceramic slurry using manual stirring and a vortex mixer at 100 rpm for 5 min. The ceramic slurry was then poured into two types of plastic molds: a round disk mold (ϕ 5 mm; h 0.7 mm) and a hollow cylinder mold (ϕ_1_ 3.5 mm; ϕ_2_ 1.5 mm; h 10 mm). The green body was slowly heated in a Nabertherm HTC-03/14 electric furnace (Lilienthal, Germany) to 600 °C for 1.5 h in air atmosphere, and the temperature was then maintained at 600 °C for 30 min. Next, the temperature was increased to 1200 °C over 30 min and maintained at that temperature for 2 h, followed by cooling to room temperature.

X-ray diffraction (XRD; Rigaku D/max VIII, Tokyo, Japan) with Cu*K*α radiation was used to characterize the phase composition of the bioceramics. Experiments testing the liquid permeability of the columnar porous bioceramics were performed with red ink, which was slowly touched on the surface of the columnar porous bioceramics. Microstructure observations of the bioceramic samples were performed with an optical camera and scanning electron microscope (SEM; Hitachi SU8000, Tokyo, Japan). The bulk density of complex geometric parts and their various porosities can be determined using the ASTM procedure C373, which is based on Archimedes’ principle. All porosity data were statistically analyzed and expressed as the mean ± standard deviation (SD).

### 4.6. Act-SIM Release Kinetics of SIM-PP NPs Loaded within Bioceramic Samples

First, the disk-shaped bioceramic samples were combined with SIM-PP NPs by dropping 20 μL onto the disks, which were placed in a fume hood to air dry. The total amount of simvastatin each disk-shaped bioceramic carried was 2.5 μg (40 μL of SIM-PP NPs colloid in disk bioceramic; SIM-PP 2.5/BC_disk_) or 5.0 μg (80 μL of SIM-PP NPs colloid in disk bioceramic; SIM-PP 5/BC_disk_). The hollow cylinder bioceramic samples were combined with SIM-PP NPs by dropping 80 μL into the inner hollow cylinder, and the bioceramics were placed in a fume hood to air dry. The total amount of simvastatin each hollow cylinder bioceramic carried was 5.0 μg (80 μL of SIM-PP NPs colloid in hollow cylinder bioceramic; SIM-PP 5/BC_cylinder_) or 10.0 μg (160 μL of SIM-PP NPs colloid in hollow cylinder bioceramic; SIM-PP 10/BC_cylinder_).

The act-SIM release characteristics of the bioceramic scaffolds with SIM-PP NPs were analyzed according to the procedure outlined in a previous study [22]. Briefly, five bioceramic scaffolds with SIM-PP NPs were resuspended in 3 mL of pH 7.4 PBS buffer. These samples were positioned in a shaker with constant agitation at 60 rpm and stored in an incubator at 37 °C. At predetermined time points (1, 2, 3, 4, 5, 6, 7, 8, 9 and 10 days), the release medium was collected, and 2 mL of the supernatant was withdrawn and replaced with 2 mL of fresh PBS buffer. The release media were then filtered through a 0.45-mm syringe filter, and HPLC (Lachrom Elite high-performance liquid chromatograph, Hitachi) analysis was conducted. The HPLC method followed the procedure outlined in our previous study [20]. Briefly, the mobile phase consisted of 1.0% acetic acid in MeCN/methanol/water (45/30/25), and the flow rate was set at 1 mL/min. The column effluent was detected at 227 nm with a UV/VIS detector. Separation was achieved using an Inertsil^®^ C18 (250 mm × 4.6 mm, 5 μm) analytical column connected to an Inertsil^®^ C18 (50 mm × 4.6 mm, 5 μm) guard column. The column temperature was 40 °C.

### 4.7. Alkaline Phosphatase Activity (ALP) and Mineralization Assay

The osteogenic effects of 0.5 µM act-SIM and the 0.5 µM act-SIM released from bioceramic scaffolds with SIM-PP NPs on BMSCs were evaluated using an ALP activity assay. BMSCs cloned from Balb/C mice, purchased from American Type Culture Collection (ATCC), were and maintained in Dulbecco’s modified Eagle’s medium (DMEM) with 10% fetal bovine serum (Gibco, BRL, Bethesda, MD, USA), 50 mg/mL sodium ascorbate, 50 units/mL penicillin and 50 μg/mL streptomycin (Gibco BRL, Bethesda, MD, USA). The BMSCs were culturedin 48-well plates (5000 cells/well) and then left untreated (OIM group) or treated with 0.5 μM act-SIM or treated as indicated with 0.5 μM act-SIM from bioceramic scaffolds with SIM-PP NPs for 5 days in bone medium [19,34], which contained DMEM, 10 mM NEAA, 10% FBS, 0.01% vitamin C, 50 g/mL streptomycin and 50 units/mL penicillin. The medium was then replaced with osteoinduction medium (OIM) consisting of low-glucose DMEM, 10% fetal bovine serum, 2.2 mg/mL sodium bicarbonate, 100 nM dexamethasone, 0.1 mM l-ascorbic acid-2-phosphate, 10 mM b-glycerophosphate, and streptomycin (10,000 U/mL)/ 0.5% penicillin (10,000 U/mL) to induce osteogenic differentiation for an additional 3 and 5 days. The ALP activity of BMSCs were quantitatively evaluated using a Phospha-Light system (Applied Biosystems, Waltham, MA, USA) with a microplate luminometer (Bio-Rad Laboratories Inc., Hercules, CA, USA) at 405 nm. All experiments were repeated at least three times, and six wells were analyzed per experiment.

### 4.8. Critical-Sized Calvarial Defects in Rats

All animals were housed at the Kaohsiung Medical University Animal Research Center, and the experimental procedures were approved by the Institutional Animal Care and Use Committee (Approval date: 22 August 2011; Approval No: 100054). A total of 20 male Sprague-Dawley rats (age 8 weeks, weight 200–350 g) were used in this study. The critical-sized calvarial defects (5 mm in width) were prepared according to our previous study [28]. The rats were anesthetized by intraperitoneal injection of xylazine hydrochloride (Bayer HealthCare; 12 mg/100 g body weight) in combination with ketamine (Parke-Davis; 10 mg/100 g body weight). The hair over the calvarium was shaved with a depilator and cleaned. An incision of midline calvarial was made. A critical-sized 5-mm hole was drilled to penetrate through the calvarial bone avoiding damaging the dura mater by using a trephine burr with constant phosphate-buffered saline (PBS) irrigation. Bone graft substitutes were then implanted into the calvarial defect sites. The wounds were sutured after implantation with 5–0 nylon sutures.

All rats with calvarial defects were randomized into four groups (5 rats/group) as follows: control group (Control), the calvarial defect without treatment; BC_disk_ group, the calvarial defect was treated with a bioceramic bone graft alone; SIM-PP 2.5/BC_disk_ group, the calvarial defect was treated with a disk bioceramic with 2.5 μg of simvastatin in SIM-PP NPs; and SIM-PP 5/BC_disk_ group, the calvarial defect was treated with a disk bioceramic with 5 μg of simvastatin in SIM-PP NPs. The rats were euthanized at 8 weeks post implantation. The calvarial specimens were harvested and fixed in 4% formaldehyde at 4 °C for 24 h, then analyzed for histomorphology.

### 4.9. Large Radial Bone Defect in Rabbits

All animals were housed at the Kaohsiung Medical University Animal Research Center, and the Institutional Animal Care and Use Committee approved the experimental procedures (Approval date: 22 August 2011; Approval No: 100054). A total of 9 male New Zealand white rabbits (weight 2.5–3 kg) were used in this study. The rabbits were first anesthetized with ketamine (45 mg/kg) and xylazine (5 mg/kg). The periosteum of the forelimb was cleared, and a 1-cm bone fragment was cut off at the center of the radius to cause the bone defect. All rabbits with the radial bone defect were randomized into three groups (3 rabbits/group) as follows: in the BC_cylinder_ group, the radial bone defect was treated with a bioceramic bone graft alone; in the SIM-PP 5/BC_cylinder_ group, the radial bone defect was treated with a bioceramic encapsulating 80 μL of SIM/PP containing 5 μg of SIM; and in the SIM-PP 10/BC_cylinder_ group, the radial bone defect was treated with a bioceramic encapsulating 160 μL of SIM/PP containing 10 μg of SIM. The rabbits were sacrificed 10 weeks after implantation. The radial bones were harvested for further analysis.

### 4.10. Soft X-ray Observation

The femur fractures were radiographically examined using soft X-rays (SOFTEX, Model M-100, Austin, TX, USA) at 0, 8, and 10 weeks after the operation at 2 mA and 43 kVP for 1.5 s. The appropriate magnification was applied throughout the observation period, and the resultant micrographs were compared among the controls and with all scaffolds.

### 4.11. Histomorphological Analysis and Immunohistochemical Staining of Bone Tissue

Quantitative histomorphological and immunohistochemical analyses were used to investigate the microstructure changes in bone tissue. All bone tissue samples were fixed in 4% formaldehyde at 4 °C for 24 h, following decalcified using 0.5 M EDTA in double-distilled water and then embedded in paraffin wax, and 5-μm sections were prepared prior to hematoxylin-eosin (H&E) and immunohistochemistry (IHC) staining. These sections were routinely stained with hematoxylin-eosin. At a magnification of 40×, we defined the callused area for counting as the 1-mm regions proximal and distal to the bone graft ends. For the quantification of new bone formation, the area of callus formation around the graft bone was measured by using Image-Pro Plus 5.0 software (Media Cybernetics Inc., Rockville, MD, USA). The percentage of new bone matrix formation within the callus was calculated and compared with that in the control group. The expression of von Willebrand factor (vWF) was determined by immunohistochemical (IHC) staining. Bone sections were treated with 0.15 mg/L trypsin in PBS at pH 7.8 for 9 min, and then the sections were incubated with PBS containing 2% BSA for 30 min at room temperature to block nonspecific binding. The sections were incubated with a 1:300 dilution of polyclonal rabbit anti-human vWF antibody (Chemicon International Inc., Temecula, CA, USA) at 4 °C. Secondary antibody of goat anti-rabbit biotinylated immunoglobulin (DakoCytomation, Copenhagen, Denmark) was used at a 1:300 dilutions for 60 min at 37 °C. An avidin-biotin-peroxidase complex kit (Vector Laboratories, Burlingame, CA, USA) was used at a 1:300 dilutions at 37 °C for 60 min. Peroxidase activity was detected by 0.4 mg/L 3,3’-diaminobenzidine in PBS with 0.12% H_2_O_2_, and then the sections were counterstained with hematoxylin.

### 4.12. Statistical Analysis

The data from representative experiments are shown and expressed as the mean ± standard error of the mean. Statistical significance was evaluated by one-way analysis of variance. The multiple comparisons were performed using Scheffe’s method. *p* = 0.05 was considered statistically significant.

## 5. Conclusions

The porous bioceramic (BC) scaffold combined with SIM-PP NPs with sustained-release of simvastatin produced both osteoinductive and osteoconductive effects that accelerated bone regeneration. The encapsulation efficiency of SIM in SIM-PP NPs was approximately 33.6 ± 3.5% in this study. However, the less than 200 nm average sizes of SIM-PP NPs can easy penetrate into the micropores of the porous bioceramic, and the combination product of the SIM-PP NPs/BC can provide sustained release of act-SIM. In addition, the act-SIM released from SIM-PP NPs/BC has positive osteogenesis effects due to changes in ALP activity. Autogenous bone-graft substitutes can provide osteogenic, osteoinductive, and osteoconductive properties. This study concludes that the combination product of a porous BC scaffold combined with SIM-PP NPs can improve the healing of bone defects in a rabbit radial endochondral ossification bone defect model and an SD rat calvarial intramembranous defect model. Therefore, optimal bioceramics containing SIM-PP NPs may be appropriate for clinical use as a synthetic alternative to autologous bone grafting that can overcome the problem of determining the clinical dosage of simvastatin drugs to promote bone healing.

## Figures and Tables

**Figure 1 ijms-19-04099-f001:**
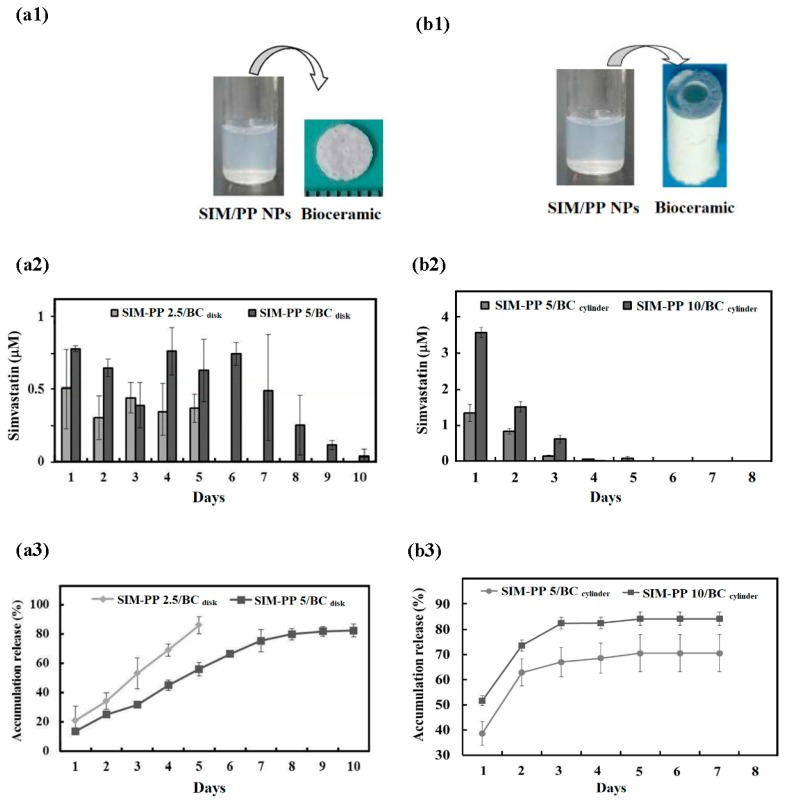
Two types of porous bioceramic samples are shown below: one is disk mold (**a1**, ϕ 5 mm; h 0.7 mm) and the other is hollow cylinder mold (**b1**, ϕ_1_ 3.5 mm; ϕ_2_ 1.5 mm; h 10 mm), these are all combined with SIM-PP nanoparticles and promote bone growth. Daily release profiles of the active form of simvastatin (act-SIM) measured by high-pressure liquid chromatography at 37 °C for 7–10 days. (**a2**) SIM-PP NPs loaded within disk bioceramic samples (ϕ 5 mm; h 0.7 mm; BC_disk_). (**b2**) SIM-PP NPs loaded within hollow cylinder bioceramic samples (ϕ_1_ 3.5 mm; ϕ_2_ 1.5 mm; h 10 mm). (**a3**,**b3**) Average percentages of act-SIM cumulative release. Notes: SIM-PP NP 2.5/BC_disk_: the 2.5 μg simvastatin loaded with disk bioceramic; SIM-PP NP 5/BC_disk_: the 5.0 μg simvastatin loaded with disk bioceramic; SIM-PP NP 5/BC_cylinder_: the 5.0 μg simvastatin loaded with hollow cylinder bioceramic; SIM-PP NP 10/BC_cylinder_: the 10.0 μg simvastatin loaded with hollow cylinder bioceramic.

**Figure 2 ijms-19-04099-f002:**
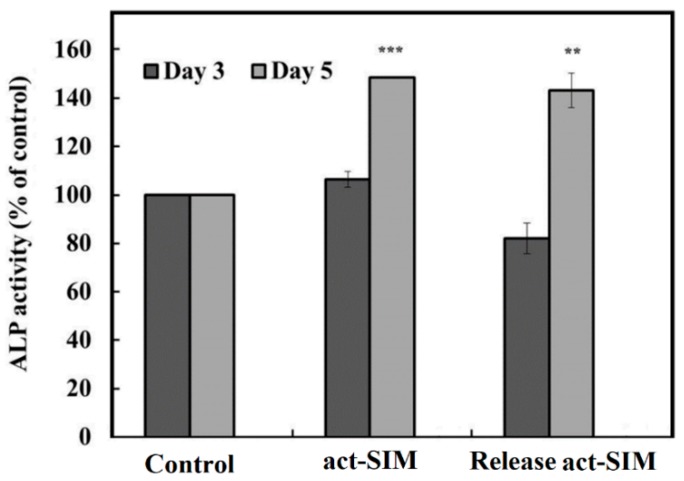
Alkaline phosphatase (ALP) activity evaluated in D1 cells. D1 cells were treated with 0.5 µM active form of simvastatin (act-SIM) in bone medium or treated with 0.5 μM act-SIM released from SIM-PP NPs loaded within bioceramic samples in bone medium for 5 days. The cells were further cultured in osteoinduction medium (OIM) for another 3 and 5 days. The control groups were not treated with act-SIM. Note: *** *p* < 0.001 and ** *p* < 0.01 com pare to day 5 Control.

**Figure 3 ijms-19-04099-f003:**
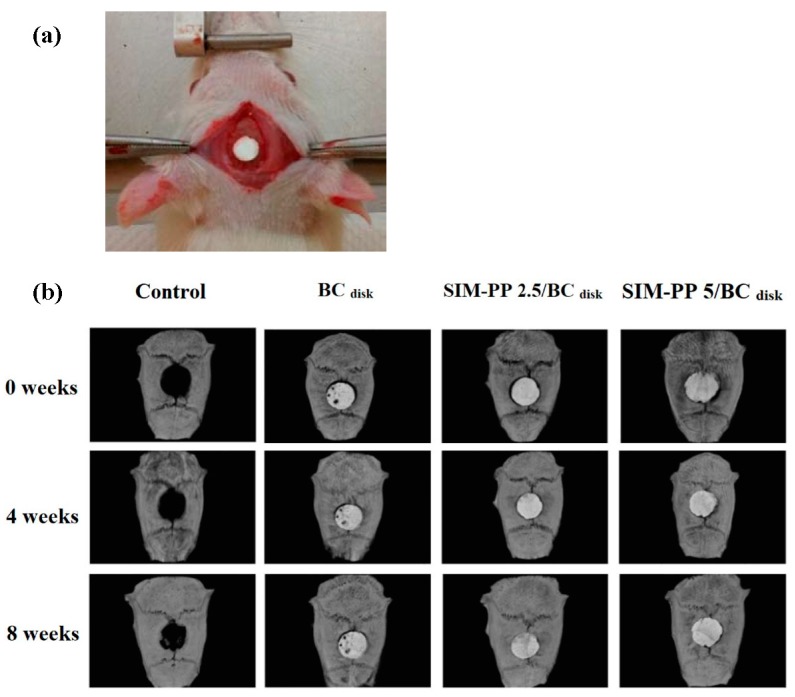
Representative calvarial bone defects of the rat model are shown in experimental photograph (**a**). The radiography study observed (**b**) calvarial bone defects of the rat model at 0, 4 and 8 weeks after implantation of 2.5 μmol of SIM and 5.0 μmol of SIM in SIM-PP/BC_disk_ samples (ϕ 5 mm; h 0.7 mm). Notes: calvarial bone defects only were used as controls.

**Figure 4 ijms-19-04099-f004:**
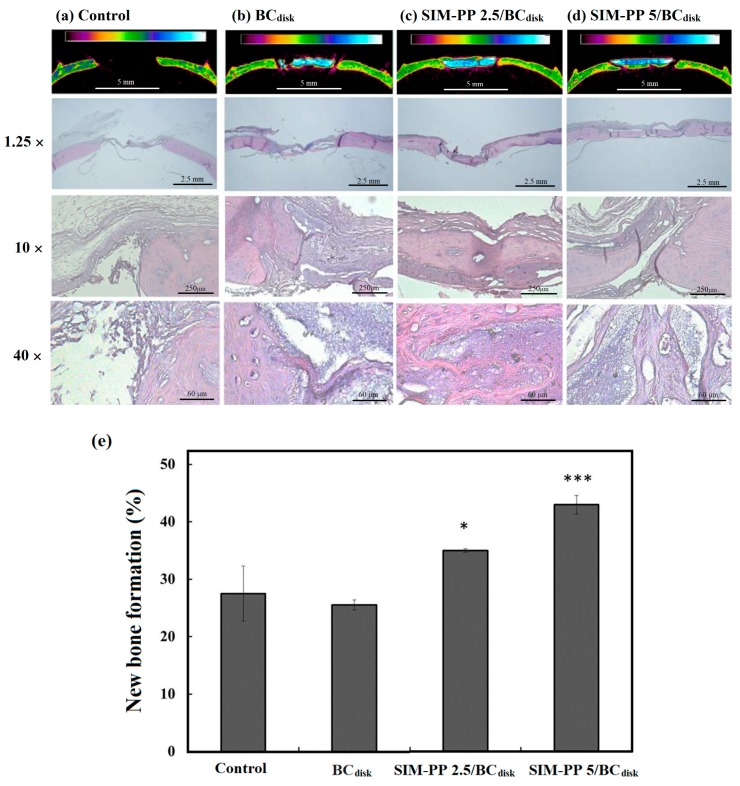
Typical Hounsfield unit (HU) calibration of micro-CT images and histological study of calvarial bone defects in the rat model only (**a**) and 8 weeks after implantation of BC_disk_ (ϕ 5 mm; h 0.7 mm) (**b**), 2.5 μmol of SIM in SIM-PP/BC_disk_ (**c**), or 5.0 μmol of SIM in SIM-PP/BC_disk_ (**d**). Callus quantification study of these calvarial bone defects groups (**e**). Notes: Non-union defects implanted without bone graft substitutes were used as controls; *N* = 5/group. * *p* < 0.05 and *** *p* < 0.001 compare control.

**Figure 5 ijms-19-04099-f005:**
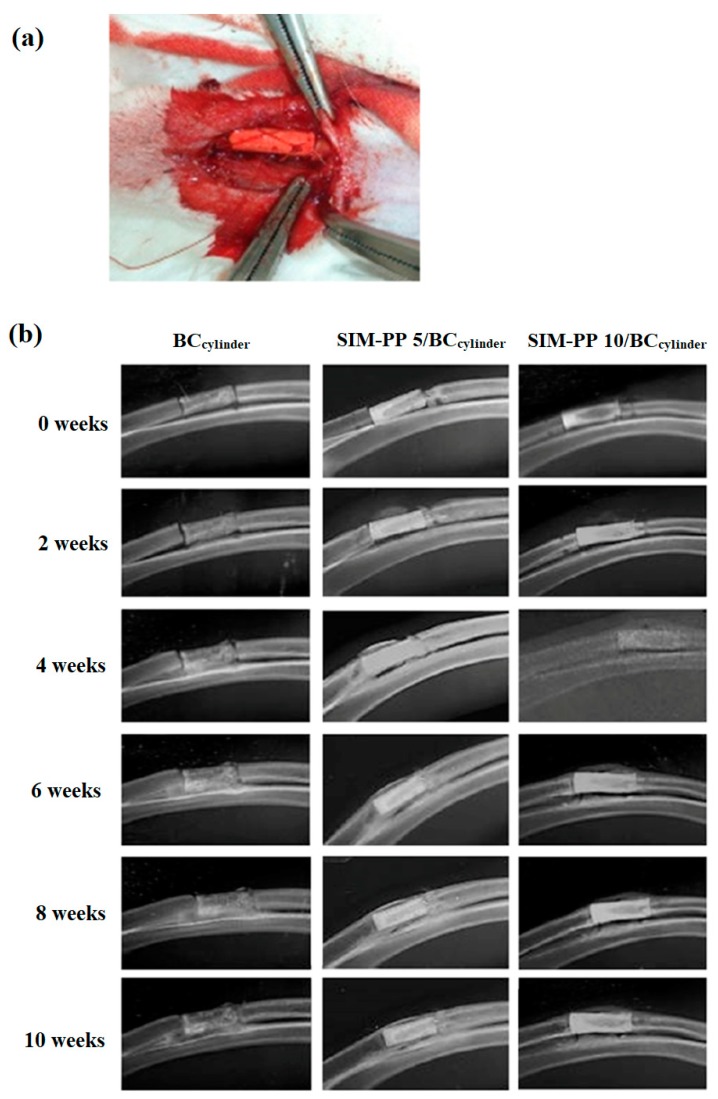
A representative radial bone defect rabbit model is shown in experimental photograph (**a**). The radiography study was performed to evaluate (**b**) the 10-mm nonunion radial bone defect rabbit model at 0, 2, 4, 6, 8 and 10 weeks after implantation of BC_cylinder_ samples (ϕ_1_ 3.5 mm; ϕ_2_ 1.5 mm; h 10 mm), 5.0 μmol of SIM in SIM-PP/BC_cylinder_ and 10.0 μmol of SIM in SIM-PP/BC_cylinder_.

**Figure 6 ijms-19-04099-f006:**
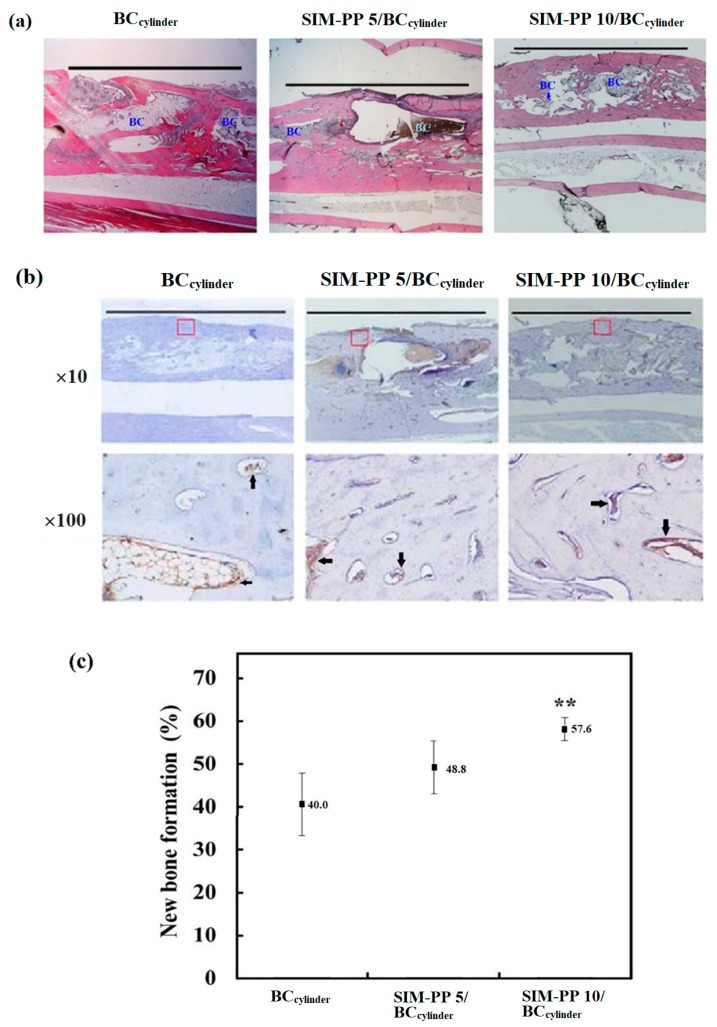
The SIM-PP 10/BC_cylinder_ group had the highest new bone formation. H&E staining (**a**) and IHC staining of von Willebrand factor (VWF) (**b**) of the 10-mm nonunion radial bone defect rabbit model at 10 weeks after implantation of BC_cylinder_ samples (ϕ_1_ 3.5 mm; ϕ_2_ 1.5 mm; h 10 mm), 5.0 μmol of SIM in SIM-PP/BC_cylinder_ and 10.0 μmol of SIM in SIM-PP/BC_cylinder_. BC: bioceramic. Arrows indicate the VWF staining. Quantification of new bone formation of these 10-mm nonunion radial bone defect rabbit groups (**c**). (*N* = 3; ** *p* < 0.01 compare BC_cylinder_). Scale bars = 10 mm.

**Table 1 ijms-19-04099-t001:** The encapsulation efficiency, loading efficiency and average size of the nanoparticles encapsulating simvastatin (SIM) (SIM-PP NPs).

Encapsulation Efficiency (EE%)	Loading Efficiency (LE%)	Mean Particle Diameter (nm)	Zeta Potential (mv)
33.6 ± 3.5	3.2	120.3 ± 8.5	−32.7 ± 3.1

Note: EE% is encapsulation efficiency percentage and LE is drug loading amount of per milligrams.

**Table 2 ijms-19-04099-t002:** The phase ratio of HAp/β-TCP, bulk density and porosity of the bioceramics after sintering at 1200 °C for 2 h.

Phase Ratio of HAp/β-TCP	Porosity (%)	Bulk Density (g/cm^3^)
50/50	56.29 ± 0.55	~1.50 ± 0.01

Note: Theoretical density (50% HAp + 50% β-TCP) = 3.11 (g/cm^3^).

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
