# Peer review of "Combination of a Bioceramic Scaffold and Simvastatin Nanoparticles as a Synthetic Alternative to Autologous Bone Grafting"

_ijms, 2018, doi:10.3390/ijms19124099_

Reviewer 1 Report

I have reviewed the manuscript “Combination of A Bioceramic Scaffold and Simvastatin Nanoparticles as A Synthetic Alternative to Autologous Bone Grafting” and found this work interesting and fit well with in the scope of this journal. Authors attempted to investigate the drug-containing bone grafts were developed from poly(lactic acid-coglycolic acid)-polyethylene glycol (PLGA-PEG) nanoparticles encapsulating simvastatin (SIM-PP NPs) loaded within an appropriately mechanical bioceramic scaffolds. However, this manuscript needs major revision and I would like to suggest following corrections before considering this work for the publication.

General comments

There are a huge number of language errors (I have mentioned a few in my comments); however, it needs careful proofreading or language editing.

Abstract

Line 22-23: “In addition to statins……” is not clear; should be rephrased

Number of specimens should be mentioned. Key details for methodology and results are deficit in the abstract.

Methodology and results

Both sections are well detailed and well-presented

Discussion of results section;

There is need of improvements; it should start with the brief description of the purpose and outcome of study followed by further discussion. Although authors justified methods and parameters, the discussion lack comparing its own results with previous studies in context. Authors may consider adding more relevant studies. And limitations of the study.

Conclusions

The section has too much details; I suggest to brief it and describe precise and to the point results. Any extra description should be moved to above section instead of conclusion. 

Author Response

Reviewer 1:

1.     Abstract: Line 22-23: “In addition to statins……” is not clear; should be rephrased.

Reply: We have rephrased the sentence in Abstract (page 2, line 4).

2.     Number of specimens should be mentioned. Key details for methodology and results are deficit in the abstract.

Reply: We have added the number of specimens (page 2, line 15) and rephrased the methodology and results in Abstract.

3.     Discussion of results section: There is need of improvements; it should start with the brief description of the purpose and outcome of study followed by further discussion. Although authors justified methods and parameters, the discussion lack comparing its own results with previous studies in context. Authors may consider adding more relevant studies. And limitations of the study.

Reply: We have edited the Discussion (Page 15~19)

 4.     Conclusions: The section has too much details; I suggest to brief it and describe precise and to the point results. Any extra description should be moved to above section instead of conclusion.

Reply: We have re-edited the Conclusion (Page 27)

Reviewer 2 Report

This paper presents a product that combines a drug-containing bone graft based on a porous bioceramic with simvastatin nanoparticles. The main idea is to enhance bone healing. Authors have already proved that the local delivery of controlled-release simvastatin/PLGA/HAp microspheres enhances bone repair. In this paper, they perform in vivo animal studies for studying the healing of non-union bone defects of radial bones and calvarial bones.
As the synthesis of the copolymer has been already published, my main concerns are related to the in vivo experiments. The histological analysis is very, and the quality of the images and results provided do not show the significant differences that the authors suggest. First, authors must provide a more detailed 2.11 section explaining the quantification methods and including more supplementary information about these procedures. Besides, authors need to address several other points before considering the paper suitable for publication:
- In the abstract, Lines 23-24 and 36 should be re-written as it is not clear the sentence.
- What are the differences in porosity between the authors' method and other standard techniques in the field of tissue engineering to obtain porous scaffolds?
- Figure 2. ALP activity is not the best technique to ensure cell viability in the scaffold. Authors' needs to provide more information that show the proliferation/cell death in the different specimens. This is important as the differences between day 3 and day 5 do not follow a clear trend. While day 3 is negative (cell activity decreasing), we observe that day 5 is just the opposite, ... This could be a cytotoxicity effect and authors must prove that there is no such as effect.
- Figure 4 shows a one-way ANOVA with 4 groups but 3 samples per group. However, the authors mentioned n = 3-5, what does it mean? Did they use a different number of samples per group? Can also explain the no variability in the SIM-PP 2.5/BC group? Also, how can be higher the value of the quantification in the group Control than in the group "BC disk"?
- Figure 6.a and .b, please, provide images with higher quality as it is not possible to evaluate the histological section by this reviewer. In the figure, please, label the different materials, tissues, elements, and so on.
- The VWF stainings provided are not clear because of the pictures. Endothelial cells do not seem to be present. Please provide more information.
- There is no information about this procedure: "Quantification of the H&E results showed that the SIM- PP 10/BC cylinder group had the highest new bone formation (Fig. 6c)". It was based just in the picture showed? Then, i totally disagree with this conclusion.
- In Figure 6.c, remove the barplot and show the real number of the 3 values. Please provide information about the normality and the correctness of the test applied as it is not clear in the figure caption. What is the p-value showing in the picture? BC vs. BC cylinder? Did the authors perform the correction because of the multiple comparison tests? Show the exat p-values of the mentioned statistical analyses.
- Line 494. What is a "Typical histological study?" and "immunohistochemical study" of what? The captions are essential to understanding if there are some differences in the different groups. However, the caption is completely useless for this aim.
- Line 489. This sentence could not be verified with the images provided. No endothelial cells seem to be present in the scaffold.
- Lastly, the English need considerable editing before being published.

Author Response

Reviewer 2:

    1. In the abstract, Lines 23-24 and 36 should be re-written as it is not clear the sentence.

Reply: We have rephrased the sentence in Abstract (page 2, line 4 and line 18).

2.     What are the differences in porosity between the authors' method and other standard techniques in the field of tissue engineering to obtain porous scaffolds?

Reply: Thank you for the opportunity to give us further explanation. Many kinds of porous bioceramic fabrication methods, include particulate/salt leaching, the polymer burning-out method, gas foaming, freeze drying, fiber bonding, emulsification, phase separation/inversion, and so on, which are difficult to control the pore shape, architecture, porosity, or interconnectivity of the bioceramic scaffolds. However, the mechanical strength is often too low to match the compressive strength of cancellous bone (approximately 4–12 MPa). Increasing the compressive strength of scaffolds without decreasing the porosity and pore size is a challenge for researchers. These explanatory sentences have been added in Discussion section (page 16-17).

3.      Figure 2. ALP activity is not the best technique to ensure cell viability in the scaffold. Authors' needs to provide more information that show the proliferation/cell death in the different specimens. This is important as the differences between day 3 and day 5 do not follow a clear trend. While day 3 is negative (cell activity decreasing), we observe that day 5 is just the opposite, ... This could be a cytotoxicity effect and authors must prove that there is no such as effect.

Reply: The caption of Figure 2 is wrong. We have corrected the caption of Figure 2 as “ Alkaline phosphatase activity evaluated in D1 cells”. ALP is not for evaluation of cell viability. As shown in Materials and methods (page 23, line 2), the purpose of ALP is to evaluate the osteogenic effects of 0.5 µM act-SIM and the 0.5 µM act-SIM released from bioceramic scaffolds with SIM-PP NPs on BMSCs. Moreover, our previous results have demonstrated that both bioceramic scaffolds [1] and SIM-PP NPs [2] have no cytotoxicity in BMSCs.  

4.     Figure 4 shows a one-way ANOVA with 4 groups but 3 samples per group. However, the authors mentioned n=3-5, what does it mean? Did they use a different number of samples per group? Can also explain the no variability in the SIM-PP 2.5/BC group? Also, how can be higher the value of the quantification in the group Control than in the group "BC disk"?

Reply: Thank you for the correction. We have confirmed the number of specimens is N=5/group in Fig. 4. It does have the variability in the SIM-PP 2.5/BC group in Fig. 4, but the variability is very small. Because there is higher variability in control group, it is acceptable that there is higher mean value in the Control group than in the group "BC disk in Figure 4e.

5.     Figure 6. a and b, please, provide images with higher quality as it is not possible to evaluate the histological section by this reviewer. In the figure, please, label the different materials, tissues, elements, and so on.

Reply: We have provided the clear image and labeled in Figure 6. a and b.

6. The VWF staining provided are not clear because of the pictures. Endothelial cells do not seem to be present. Please provide more information.

Reply: Thank you for the comment. We have provided clear picture of Fig. 6b. The von Willebrand factor (VWF) is a secreted glycoprotein of endothelial cells, therefore, VWF staining can see the area of new blood vessel formation, but not the endothelial cells.

7.     There is no information about this procedure: "Quantification of the H&E results showed that the SIM- PP 10/BC cylinder group had the highest new bone formation (Fig. 6c)". It was based just in the picture showed? Then, I totally disagree with this conclusion.

Reply: The procedure of quantification of new bone formation is showed in Material and Methods section (page 26, line 10). For the quantification of new bone formation, the area of callus formation around the graft bone was measured using Image-Pro Plus 5.0 software (Media Cybernetics Inc., Rockville, MD, USA). The percentage of new bone matrix formation within the callus was calculated and compared with that in the control group.

8. In Figure 6.c, remove the bar plot and show the real number of the 3 values. Please provide information about the normality and the correctness of the test applied as it is not clear in the figure caption. What is the p-value showing in the picture? BC vs. BC cylinder? Did the authors perform the correction because of the multiple comparison tests? Show the exact p-values of the mentioned statistical analyses.

 Reply: We have re-moved the bar plot and showed the real number of the 3 values in Figure 6. c. We also revised the caption in the figure 6 including p-value. ** P<0.01 compare BCcylinder. We have re-checked our parametric test with SPSS (ver. 22.0) software to confirm the normality and correctness. The results showed that the data presented in this study had values > 0.05 in variance homogeneity, which means that it met the homogeneity between groups. By using Shapiro-Wilk analysis, we found that the results met the normality with two very close mean value and median value in the individual group. We thought that our statistical analysis was applicable and without against the normality and correctness in this study.

9. Line 494. What is a "Typical histological study?" and "immunohistochemical study" of what? The captions are essential to understanding if there are some differences in the different groups. However, the caption is completely useless for this aim.

Reply: We have change the caption of Figure 6 to “ The SIM-PP 10/BC cylinder group had the highest new bone formation.”.

10. Line 489. This sentence could not be verified with the images provided. No endothelial cells seem to be present in the scaffold.

Reply: Thank you for the comment. We have provided clear picture of Fig. 6b. VWF staining can see the area of new blood vessel formation, but not the endothelial cells.

11. the English need considerable editing before being published.

Reply: We have edited the English at September 14, 2018 (see attached certificate). We have also rephrased some sentence in this revised manuscript.

References:

1.    Fu, Y. C.; Chen, C. H.; Wang, C. Z.; Wang, Y. H.; Chang, J. K.; Wang, G. J.; Ho, M. L.; Wang, C. K., Preparation of porous bioceramics using reverse thermo-responsive hydrogels in combination with rhBMP-2 carriers: in vitro and in vivo evaluation. J Mech Behav Biomed Mater 2013, 27, 64-76.

2.    Wang, C. Z.; Fu, Y. C.; Jian, S. C.; Wang, Y. H.; Liu, P. L.; Ho, M. L.; Wang, C. K., Synthesis and characterization of cationic polymeric nanoparticles as simvastatin carriers for enhancing the osteogenesis of bone marrow mesenchymal stem cells. J Colloid Interface Sci 2014, 432, 190-9.

Round  2

Reviewer 2 Report

The authors addressed reviewer´s comments, therefore the paper can be accepted in the present form.